# AquaCropPlotter: A Shiny app for visualizing and analyzing AquaCrop simulation results

**Nattapong Sanguankiattichai**[1,2,3]*, **Andrea Setti**[1], **Jorge Alvar-Beltrán**[1], **Maher Salman**[1], **Dirk Raes**[4], **Riccardo Soldan**[1]*

**1** Food and Agriculture Organization (FAO) of the United Nations, Rome, Italy, **2** Department of Biology, University of Oxford, Oxford, United Kingdom, **3** Department of Microbiology, Faculty of Science, Mahidol University, Bangkok, Thailand, **4** Department of Earth and Environmental Sciences, KU Leuven, Leuven, Belgium

* nattapong.san@mahidol.ac.th (NS); riccardo.soldan@fao.org (RS)

## Abstract

AquaCrop, a crop growth model developed by the Food and Agriculture Organization of the United Nations (FAO), can be used as a planning tool to assist management decisions in both irrigated and rainfed agriculture. AquaCrop simulates the yield of crops in response to water, particularly in conditions where water is a key limiting factor. AquaCrop balances accuracy, simplicity and robustness. However, visualization and analysis of its outputs can be a bottleneck since data generated from simulation runs are stored in a number of text files that contain a large amount of information. Processing these data can be challenging, especially at the high throughput commonly needed when comparing multiple models or assessing combinations of different factors. To address this limitation, we developed AquaCropPlotter, an R Shiny application designed to streamline the processing, visualization and analysis of AquaCrop outputs. The workflow of AquaCropPlotter starts with uploading a batch of all the AquaCrop output files selected by the user, followed by automated processing of the data into clean structured tables, which can then be explored with flexible visualization functionalities and simple statistical analysis tools. The resulting data tables, plots and analysis outputs can be readily exported for use in reports or further analysis. AquaCropPlotter is an open-source software available as a web application, locally installed application and R package. Its intuitive interface enables users to easily gain insights from AquaCrop simulation results without requiring programming expertise. As a case study, we applied AquaCropPlotter to analyze AquaCrop simulations conducted in the Republic of Moldova. Overall, AquaCropPlotter aims to facilitate broader utilization of AquaCrop and support its application across agricultural research and practice.

**Data availability statement:** AquaCropPlotter is an open-source software, available both as an online web interface and an offline local installation. The online web application is freely accessible on the shinyapps.io server https://foodandagricultureorganization.shinyapps.io/AquaCropPlotter. For Windows users, AquaCropPlotter can also be installed locally as a stand-alone app using an installer provided at https://github.com/Risk-Team/AquaCropPlotter/releases. AquaCropPlotter is also available as an R package that can be installed from Github (Risk-Team/ AquaCropPlotter/AquaCropPlotter) using the devtools package. The full AquaCropPlotter user guide, with instructions and case studies, is available at FAO Knowledge Repository (DOI: 10.4060/cd0086en). Example AquaCrop output data that can be used with AquaCropPlotter are available at Mendeley Data (DOI: 10.17632/ynz4zt54r4.1). Example AquaCrop output data of maize in the Republic of Moldova are available at Mendeley Data (DOI: 10.17632/xkzsrwxwbs.1).

**Funding:** The author(s) received no specific funding for this work.

**Competing interests:** The authors have declared that no competing interests exist.

## Introduction

Food security and sustainable use of natural resources are among the most pressing global challenges, which continue to be exacerbated by climate variability and weather extremes, water scarcity and the increasing demands of a growing population. With agricultural productivity playing an important role in sustaining global food supply, crop models can be valuable tools for the effort to address these issues. By enabling the simulation of crop growth and yield under different environmental conditions and management scenarios, crop models provide insights that facilitate risk assessment, exploration of adaptation solutions, and optimization of resource use [1].

AquaCrop, developed by the Food and Agriculture Organization of the United Nations (FAO), is a crop-water productivity model designed to simulate the crop biomass and yield in response to water availability, consumption and agronomic management, while integrating the concepts of plant physiology, soil water dynamics and salt balance, among other abiotic stresses [2–7]. AquaCrop simulations use several input data including climate, crop, soil, and management parameters [2–7].

- **Climate:** Input data include daily minimum and maximum air temperature, reference evapotranspiration (ETo), calculated using the FAO Penman-Monteith method, rainfall, and annual atmospheric $CO_2$ concentration, which are used to adjust water productivity (WP*).

- **Crop:** Input data are organized in a crop calendar, specifying sowing or planting dates (fixed or triggered by climatic thresholds), and crop characteristics, which include conservative parameters (stable across environments), cultivar-specific traits, and non-conservative parameters (sensitive to local management or environmental conditions). FAO provides calibrated defaults values for major crops, while users may generate new crop files when needed.

- **Soil:** Input data describe up to five soil horizons, characterized by volumetric water content at saturation, field capacity, wilting point, hydraulic conductivity, presence of gravel, soil penetrability, and groundwater table depth and salinity.

- **Management practice:** Input data are divided into field and irrigation management. Field management includes fertility level, mulching, tillage, bunding, weed control, and harvest rules for multi-cut crops, with fertility status affecting water productivity, canopy development, and senescence. Irrigation management involves defining rainfed or irrigated systems, method (sprinkler, drip, or surface), fraction of surface wetted, water quality, and application scheduling. The model also supports scenario testing, such as deficit irrigation strategies [7].

Together, these inputs define the crop-soil-atmosphere-management system that AquaCrop simulates, allowing consistent yet flexible application across a range of environments. However, modelling with AquaCrop still has limitations in some cases. For example, the model does not include modules for pests and diseases, whose impacts are highly variable and difficult to parameterize, especially under changing climate conditions [8]. In addition, nutrient limitations are only indirectly represented

through simplified estimations of deficiencies in mineral nutrients, soil organic matter, and salinity in the root zone, which restricts the model's ability to capture detailed nutrient dynamics [5,9].

AquaCrop simulations work in four main steps through a daily time series across the crop growing season [2–7]. First, the model estimates the development of green canopy cover (CC), the fraction of the soil surface covered by the canopy that is affected by the prevailing water, fertility, and salinity stress, which are tracked each day. Second, crop transpiration (Tr) is calculated as the product of the ETo and a crop coefficient (Kc) that depends on the simulated CC throughout the crop life cycle. Third, the above ground biomass (B) is calculated from the accumulative amount of crop transpiration (ΣTr) with a proportional factor, normalized biomass WP parameter (WP*) (Eq.1). Finally, the crop yield (Y) is determined by multiplying the accumulated above ground biomass (B) by the harvest index (HI) (Eq.2). Additionally, AquaCrop accounts for environmental stresses such as water shortage, extreme temperature, salinity, and soil fertility limitations by applying stress coefficients (Ks) that reduce canopy cover (CC) expansion, crop transpiration (Tr), biomass production (B), or harvest index (HI).

$$B = \Sigma Tr \times WP^* \tag{Eq. 1}$$

$$Y = HI \times B \tag{Eq. 2}$$

The outputs of AquaCrop provide insights into both crop performance and the soil water balance under varying climatic and management conditions. Key results include time series of canopy cover, transpiration, biomass, and yield, which trace crop development during the growing season. In parallel, the model reports information about the water balance, such as soil moisture, evaporation, infiltration, runoff, drainage, and irrigation demand, which are critical for evaluating water productivity and management strategies. By combining crop and water balance outputs, AquaCrop supports the quantification of yield response to water, the assessment of irrigation or rainwater use efficiency, and the analysis of effects of climate variability, soil fertility, or salinity on production [7]. AquaCrop has been widely used in research, field projects and decision-making support, contributing to applications such as forecasting crop yields, assessing impacts of climate change, informing agricultural management strategies and assisting policy development across diverse regions and nations [10].

AquaCrop is available as a pre-compiled software package in two versions, standard (GUI) and stand-alone (plug-in) [7,11]. The standard version provides a graphical user interface (GUI) for running simulations manually [7], while the stand-alone (plug-in) version is designed for batch processing [11]. It runs simulations from a list of pre-defined projects and stores results in output files, useful for applications that require the evaluation of multiple iterative or parallel runs. In general, AquaCrop inputs a set of parameters specified by the users and outputs the results in a number of text files for each simulation. Oftentimes, multiple AquaCrop simulations are run over many years, in a range of locations, under different climate scenarios or with various management conditions, such as sowing date, irrigation schemes and soil management practices. Assessing combinations of these factors further multiply the number of simulations and output files. The plethora of output data generated by the model can pose a challenge for simultaneous processing and extracting insights from the data.

Since the release of AquaCrop, several other tools have been developed to complement or expand its application. AquaData and AquaGIS were developed to address the need for extensive processing of input and output files of AquaCrop simulations at scale for spatial and temporal analyses [12]. AquaCrop-OS was developed as an open-source alternative that is implemented in MATLAB and can be run on multiple operating systems [13]. AquaCropR extends the accessibility of the AquaCrop model to the open-source R programming language [14]. AquaCrop-OSPy is another open-source alternative that is implemented in Python and can be adapted to various modes of use including cloud computing [15]. Aqua-MC is a framework designed to facilitate a large number of AquaCrop runs with varying parameters in Monte

Carlo simulations, implemented in MATLAB [16]. Since the release of AquaCrop version 7, an open-source version-controlled Fortran90 code and standalone programs are available for Windows, macOS and Linux operating systems. These assets aim to serve the broader user community to efficiently exploit the capability of AquaCrop for scientific field-scale applications, global applications, practical web-based applications, or targeted user applications [17]. Although these tools can be powerful and user-friendly, they are based on coding that might preclude some users without programming background.

In this work, AquaCropPlotter was developed to provide an intuitive user interface that streamlines the processing, visualization and analysis of data from AquaCrop simulations. This tool aims to enable users to generate insights from large and complex data without requiring extensive computing skills. AquaCropPlotter facilitates the handling of AquaCrop output through automated data upload and processing, flexible data visualization and simple statistical analysis. As a case study, we applied AquaCropPlotter to analyze AquaCrop simulations conducted in the Republic of Moldova.

## Application development

AquaCropPlotter was developed in R programming language [18] using the R Shiny framework for web application development [19]. The user interface was created using additional packages including shinydashboard [20] for the overall structure of the application, shinyjs [21] for using JavaScript operations in the interactive interface, shinyBS [22] for using Bootstrap components in the interactive interface and DT [23] for interactive data table display. Data processing and analysis pipelines were created using Tidyverse packages [24] for data manipulation and calculations, and furrr [25] for parallelizing processes. Data visualization functionalities were created based on ggplot2 [26] for plotting and scales [27] for formatting scales on the plots. For the web application, rsconnect [28] was used to deploy the app in the Shinyapps.io server. For the stand-alone application in Windows, RInno [29] was used to compile the app installer. The R package was created using devtools package [30]. The full source code of AquaCropPlotter is available on Github [31].

## Access and availability

AquaCropPlotter is an open-source software, available both as an online web interface and an offline local installation. The online web application is freely accessible on the shinyapps.io server [32]. For Windows users, AquaCropPlotter can also be installed locally as a stand-alone app using an installer provided on Github [33]. AquaCropPlotter is also available as an R package that can be installed from the Github repository Risk-Team/AquaCropPlotter/AquaCropPlotter using the devtools package [30]. The full AquaCropPlotter user guide, with instructions and case studies, is available at FAO Knowledge Repository [34]. Example AquaCrop output data that can be used with AquaCropPlotter are available on Mendeley Data [35]. Example AquaCrop output data of maize in the Republic of Moldova are available on Mendeley Data [36].

## AquaCropPlotter workflow

The workflow of AquaCropPlotter consists of four steps, as summarized in Fig 1.

### Step 1: Upload data

AquaCropPlotter takes project files and output files from AquaCrop simulation as its input. Output files from both the standard "GUI" and stand-alone "Plug-in" of AquaCrop versions 6 and 7 are supported. Details of all output files and variables are provided in S1 Table.

For the standard "GUI" AquaCrop, users can select to save outputs of each simulation in 10 files, each with the file name suffix as follows.

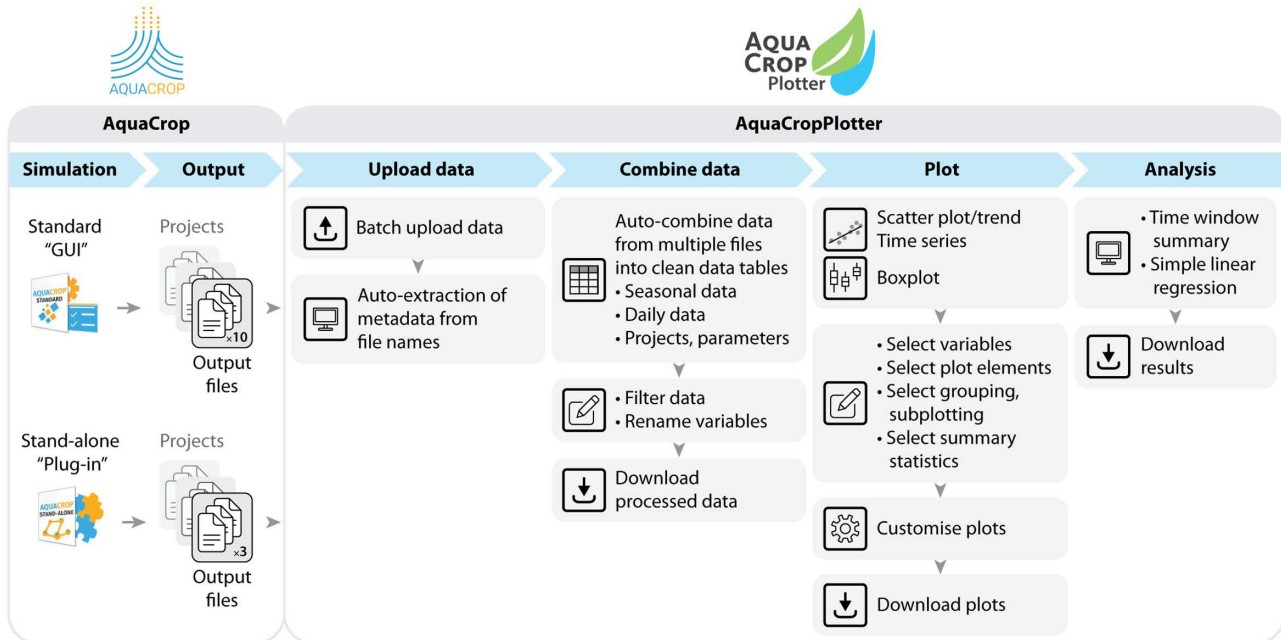

**Fig 1. Overview of AquaCropPlotter workflow.** AquaCropPlotter is designed to facilitate processing, visualization and analysis of outputs from AquaCrop simulations. The workflow consists of four main steps: upload data, combine data, plot and analysis.

## Project file

1. PRM containing information about the simulation run such as growing period, settings of parameters and input file names.

   Daily results files:

2. Clim.OUT containing variables regarding climate input parameters.

3. CompEC.OUT containing variables regarding soil salinity at various depths of the soil profile.

4. CompWC.OUT containing variables regarding soil water content at various depths of the soil profile.

5. Crop.OUT containing variables regarding crop development and production.

6. Inet.OUT containing variables regarding net irrigation requirement.

7. Prof.OUT containing variables regarding soil water content in the soil profile and root zone.

8. Salt.OUT containing variables regarding soil salinity in the soil profile and root zone.

9. Wabal.OUT containing variables regarding soil water balance.

## Seasonal results file

10. Run.OUT containing the summary of daily data in a season.

For the stand-alone "Plug-in" AquaCrop, users can save outputs in 3 files per simulation, each with the file name suffix as follows.

1. Project file (.PRM with multi-simulation project, or.PRO with single-simulation project) containing information about the simulation run such as growing period, settings of parameters and input file names.

2. Daily results files (day.OUT) containing daily breakdown of all variables including climate input parameters, crop development and production, soil water balance, net irrigation requirement, soil water content and salinity at various depths of the soil profile and root zone.

3. Seasonal results file (season.OUT) containing the summary of daily data in each season including length and dates of the simulation period; the totals for climatic, soil water and soil salinity parameters; average stresses during the growing period; final biomass production, crop yield and crop water productivity.

All output files from one or multiple simulation runs can be uploaded simultaneously for a combined analysis (Fig 2). This easy one batch upload and automated processing functionality facilitates the handling of a large amount of AquaCrop output data that has previously been one of the bottlenecks.

For each run of AquaCrop simulation, all output files have the same name prefix by default, which allows AquaCropPlotter to automatically sort and match all the files from the same simulation during upload. While processing the files, AquaCropPlotter can also extract additional metadata information from the file name, if created according to a simple file naming convention where each variable to be extracted is separated by an underscore (Fig 2). This feature allows users to provide additional metadata for each simulation that can be used in downstream analysis. For example, when different project files are made for different management practices, sowing dates, irrigation schemes, locations or climate models are analyzed together for comparison.

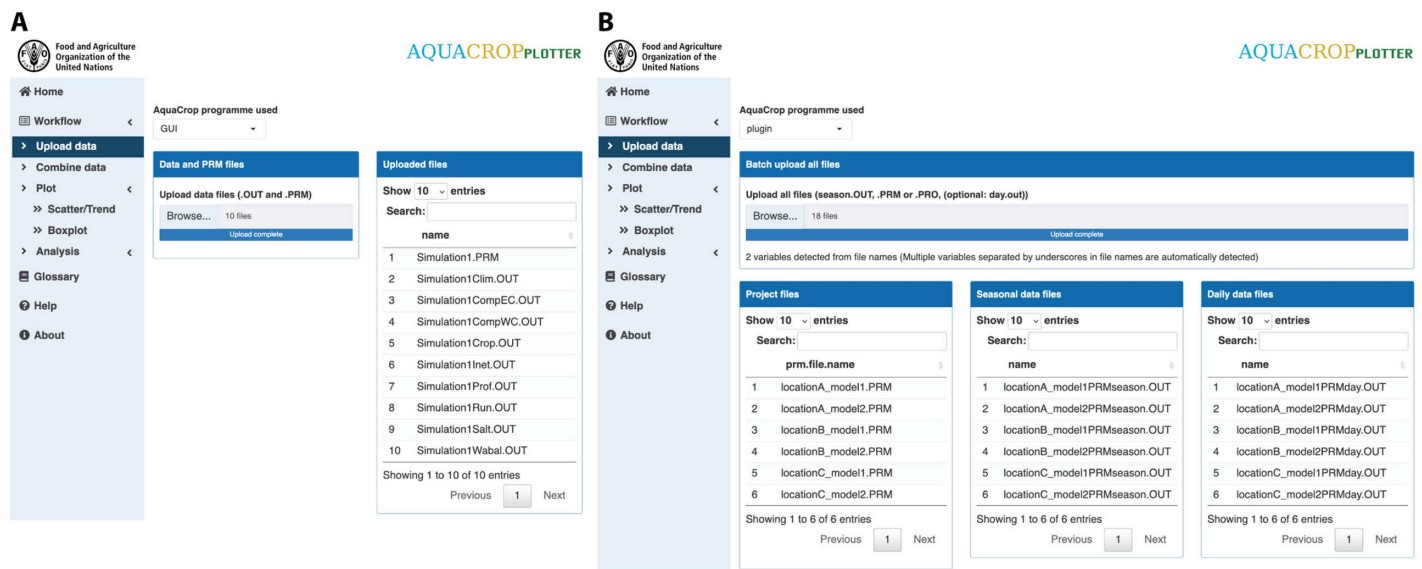

**Fig 2. AquaCropPlotter "Upload data" functionality.** The user interface for uploading AquaCrop output files, with examples of **(A)** Standard "GUI" or **(B)** Stand-alone "Plug-in" output data uploaded. Users can select files to upload in batch and the files will be automatically processed and displayed in the list.

## Step 2: Combine data

AquaCropPlotter automatically processes all the uploaded files and extracts information, such as simulation results and parameters used in the projects, which are combined into a clean data table for each type of dataset (daily, seasonal and project files) (Fig 3). These data can also be filtered or have variables renamed. The combined data are in a readily amenable format that users can export for further analysis by other software or record keeping. These datasets also serve as the foundation for the subsequent plotting and analysis steps in AquaCropPlotter.

## Step 3: Plot

AquaCropPlotter provides a flexible plotting tool for data visualization. Data can be visualized as scatter plots with depiction of relationships, trends and time series, or as boxplots (Fig 4). Users can select variables to plot and plot elements to display. For deeper insights or comparisons, grouping variables can also be selected to highlight different sets of data or split the plot into subplots. Summary statistics can also be plotted to summarize the data or improve the visualization of results. Next, AquaCropPlotter also provides extensive customization of the plot appearance and labels to suit various needs of users. Finally, the plots can be exported ready for use in reports, presentations or hands-on training.

## Step 4: Analysis

AquaCropPlotter offers two simple statistical analysis options for exploring the data (Fig 5). First, a time window summary function allows users to partition the dataset into a number of time windows, each with a user-specified time interval. Then summary statistics, including the mean, standard deviation (SD) and coefficient of variation (SD/mean), are calculated for the variable of interest within each time window and displayed in a table. This allows users to examine how a variable evolves across consecutive time windows and to identify long-term trends. For example, in a century-long simulation, one may group the data into 20-year intervals to see temporal patterns and variability over the entire time span.

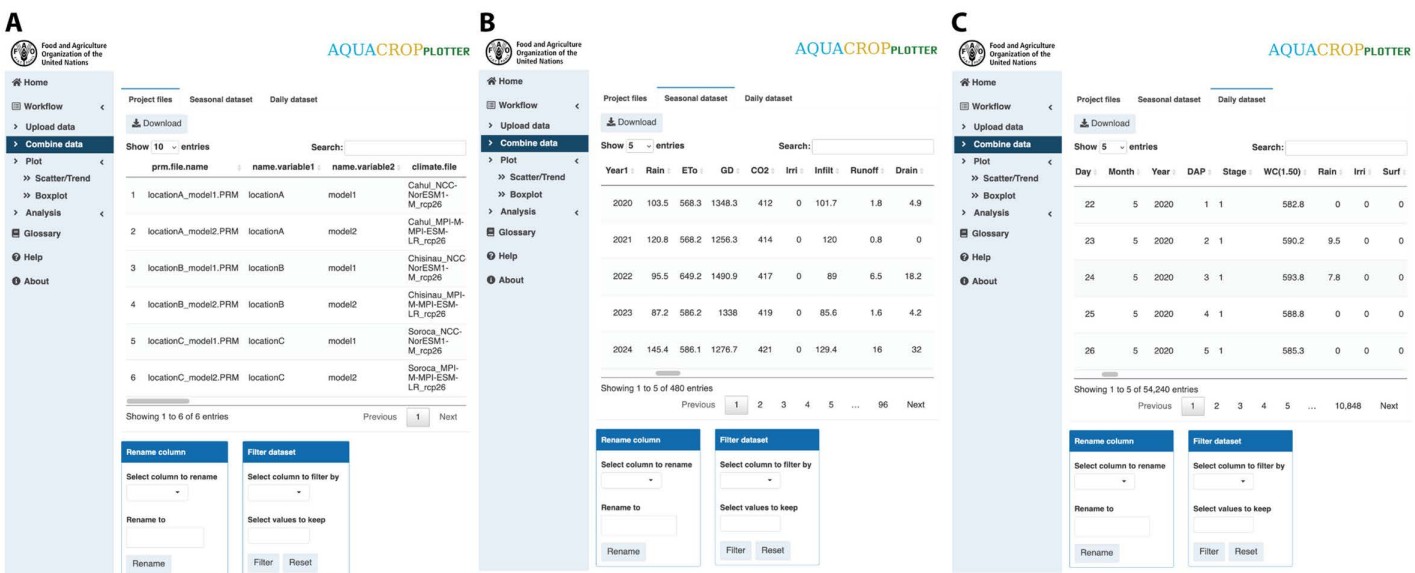

**Fig 3. AquaCropPlotter "Combine data" functionality.** The user interface for displaying combined datasets from all uploaded and processed files, with examples of **(A)** project files, **(B)** seasonal data and **(C)** daily data. Additional toolboxes at the bottom allow further data filtering and column renaming. These processed tidy datasets can also be downloaded for record or further analysis.

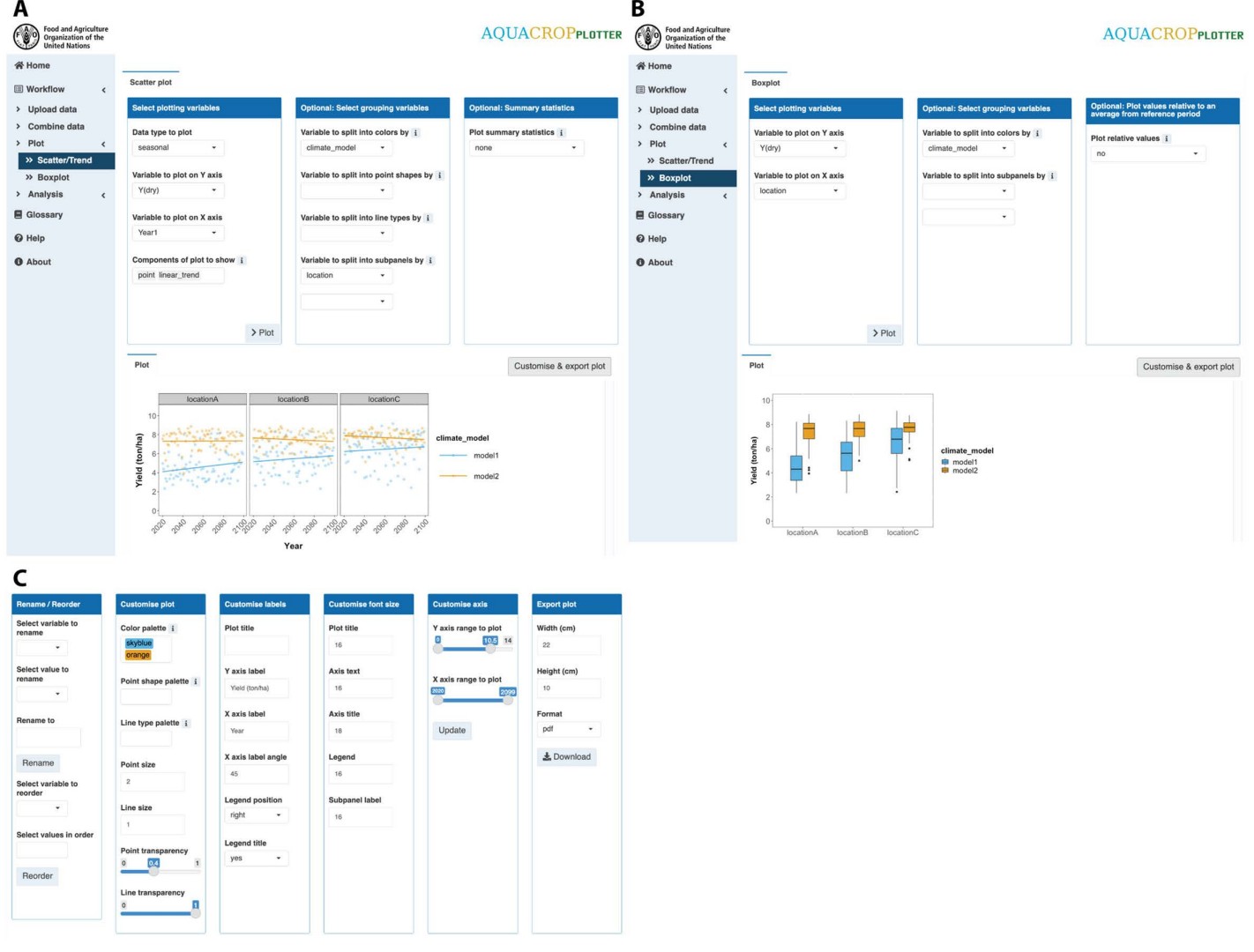

**Fig 4. AquaCropPlotter "Plot" functionality.** The user interface for visualization of data with examples of simulated maize yields over 80 years across 3 locations (Soroca, Chisinau and Cahul) in the Republic of Moldova under 2 climate scenarios (Representative Concentration Pathways – RCPs 2.6 and 8.5) plotted as **(A)** scatter plot and **(B)** boxplot. Users can select the data and variables to plot, with options to select plotting elements and groupings of data to display **(C)** Additional toolboxes allow plot customization and export.

Second, a simple linear regression analysis is available for investigating relationships between variables of interest. Users can specify the dependent and independent variables for running linear regression, which is performed using the lm() function in R. The output table will report key results including the R-squared value indicating the proportion of variance in the dependent variable explained by the model to assess how well the model fits the data, the slope coefficient indicating how much the change in the independent variable affects the dependent variable, and the p-value from a t-test to assess whether the slope is significantly different from zero. This enables a quick examination of potential linear relationships between variables of interest within the data.

For both analyses, users can further subdivide the data by user-specified grouping variables to explore the pattern within specific subgroups or compare results between groups. The analysis results displayed in a table can also be exported.

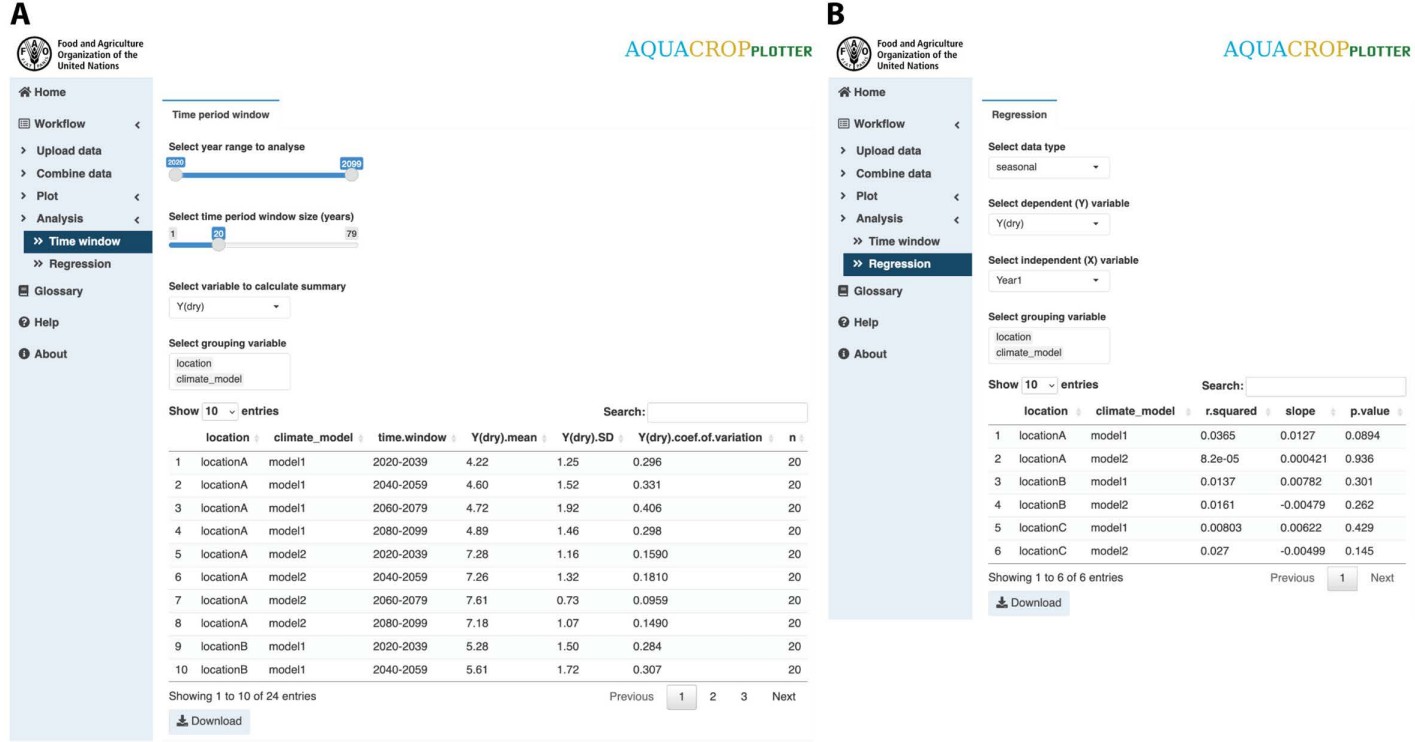

**Fig 5. AquaCropPlotter "Analysis" functionality.** The user interface for simple statistical analysis of the data, with example analyses of simulated maize yields over 80 years 3 locations (Soroca, Chisinau and Cahul) in the Republic of Moldova under 2 climate scenarios (Representative Concentration Pathways – RCPs 2.6 and 8.5), showing **(A)** time window summary of yields in 20-year time windows, **(B)** simple linear regression of yields over years. Users can select the data, variables and parameter to analyze and the results will be displayed in the table that can be downloaded.

## AquaCropPlotter case study

A recent study [37] piloted the use of AquaCropPlotter to streamline the processing and visualization of outputs from the AquaCrop stand-alone (plug-in) program. This research assessed the impacts of climate change and adaptation strategies on various crops in the Republic of Moldova. AquaCrop modeling was conducted on five crops (maize, tomatoes, sunflowers, green peas, and wheat), three locations (Soroca, Chisinau, and Cahul), three sowing dates (early, 26 Apr; reference, 11 May; and late, 26 May), two management practices (optimal and non-optimal), two climate change scenarios (Representative Concentration Pathways (RCP) 2.6 and 8.5), and three Global Climate Models (MOHC-HadGEM2-ES, MPI-M-MPI-ESM-LR, and NCC-NorESM1) over the simulation period of 90 years (2010–2100). Altogether, this study generated 648 seasonal project files (108 files for each crop), which encapsulate all 41-simulation seasonal output parameters, such as climatic, soil-water, soil-salinity, biomass production, and crop yield [11]. These output files were processed using AquaCropPlotter. Overall, the analysis of AquaCrop outputs was facilitated by AquaCropPlotter automatically processing the large dataset and allowing users to specify variables of interest (e.g., yield), define analysis time window, and group conditions for comparison (e.g., adaptation strategies). These selections were used to create plots and calculate summary statistics to provide insights for assessing the effect of each adaptation on crop yield under different conditions.

To illustrate the use case of AquaCropPlotter, here we present an example analysis of the effects of temperature stress on crop productivity under varying conditions and the impact of different sowing date adaptations, using AquaCrop simulations conducted on the medium cycle variety maize grown with non-optimal management conditions at three locations (Soroca, Chisinau, and Cahul) in the Republic of Moldova over the course of the century under the influence of two climate change scenarios (RCPs 2.6 and 8.5) [37].

In this case study, the elevated average temperatures increased the accumulation of days where the temperature exceeds the minimum temperature required for optimal growth (growing degree day, GDD), resulting in shorter crop cycle length. This decreasing trend in crop cycle length was especially pronounced under the RCP 8.5 scenario (Fig 6A), which has the highest increase in temperature during the simulated period. The delayed sowing date (26 May) also exposed the crop to higher average temperatures in the growing season, leading to shorter crop cycles compared to earlier sowing dates (Fig 6A). Linear regression analysis of crop cycle length over time supports the observed decreasing trend in all locations and sowing dates under the RCP8.5 scenario, showing significant negative slope coefficients (Fig 6B).

Temperature stress arises when the temperature falls outside the optimal range for the crop (9–30°C), leading to a reduction in stomatal transpiration, which can in turn decrease biomass accumulation. The shortening of the crop cycle can reduce the total duration of exposure to temperature extremes that causes stomatal stress, as observed in the similar decreasing trend instomatal stress days (Fig 7A), particularly for the delayed sowing date (26 May), which may partially mitigate the negative impact on productivity. Linear regression analysis of stomatal stress over time supports the observed decreasing trend in all locations and sowing dates under the RCP8.5 scenario, showing significant negative slope coefficients (Fig 7B).

However, the late sowing date (26 May) exhibited a decline in water productivity under RCP 8.5 across all locations especially from mid-century to the end of the century (Fig 8A). Using the time window summary function to calculate the mean water productivity in 20-year time windows over the century, a dip in the mean of water productivity can be observed in the late sowing date (26 May) under RCP 8.5 by the end of the century (2080–2099) (Fig 8B). This pattern reflects the compounding effects of elevated temperature and shortened crop cycles. While higher temperatures increase GDD accumulation and reduce exposure to extreme temperatures, the shorter growing period appears to limit biomass accumulation more strongly, resulting in reduced water productivity. In contrast, early sowing dates (26 April) showed more stable water productivity, particularly under RCP 2.6 (Fig 8A).

## Discussion

AquaCropPlotter was developed to facilitate the processing and analysis of data from AquaCrop simulations. Particularly, this tool addresses the challenges in the generation of insights from large and complex datasets and the accessibility for users without extensive computing skills. Therefore, AquaCropPlotter provides a simple user interface that features automated data upload and processing, flexible data visualization and simple statistical analysis.

While AquaCrop's stand-alone version 7.1 enables batch processing of multiple projects through its plug-in architecture [11], its native output remains constrained to text-based formats containing 41 parameters in multiple files. This structural limitation creates bottlenecks such as analytical inefficiency and visualization constraints, which are addressed by features of AquaCropPlotter such as automated data processing, selective parameter filtering, data aggregation frameworks and flexible visualization pipelines giving readily available plots.

The interoperability between AquaCrop and AquaCropPlotter also helps to address documented challenges, such as testing multiple adaptation solutions in large-scale agricultural modelling [38]. Current large-scale agricultural modelling is bound to limited testing of crop management practices with reduced number of factors and levels for the analysis of variance. Failing to comprehensively test various adaptation strategies may result in poor advice to policymakers and stakeholders for the development of adaptation strategies [39]. Thus, testing diverse management strategies in crop modelling provides critical insights for optimizing agricultural systems under varying environmental and socioeconomic conditions. This approach is vital for policymakers and extension services to design adaptive strategies for climate resilient farming, facilitate technology adoption, assess how practices perform across spatiotemporal scales, and use agricultural inputs more efficiently to balance productivity with sustainability. AquaCropPlotter fills this gap by enabling users to investigate multiple variables of interest from relatively large or complex analyses, thereby easing one of the major hurdles in working with the AquaCrop model.

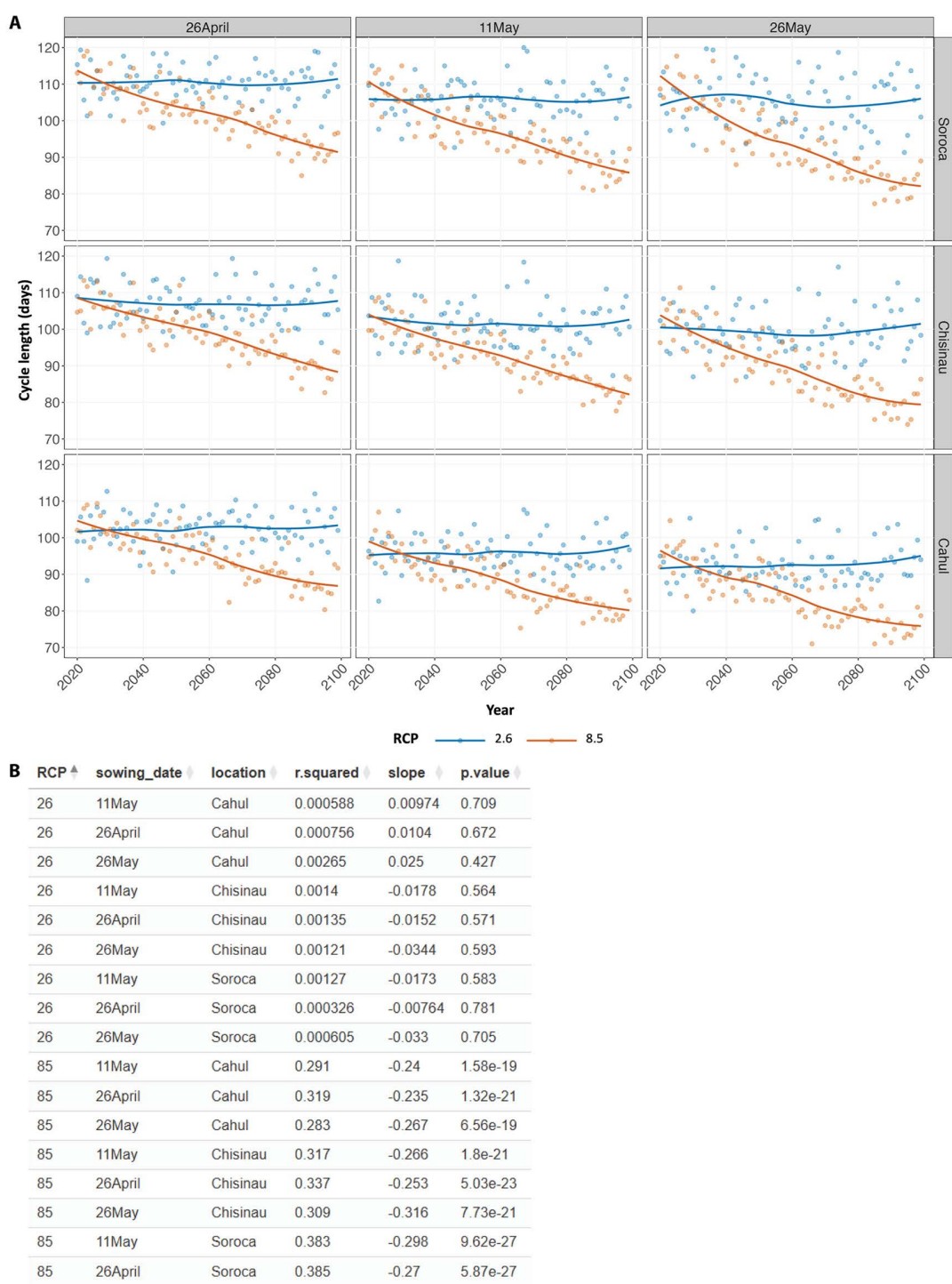

**Fig 6. Shorter crop cycle length was observed in conditions with exposure to elevated temperature. (A)** Crop cycle lengths (days) predicted from AquaCrop simulations of maize grown at three locations in the Republic of Moldova (Soroca, Chisinau, and Cahul) over the course of the century under two climate change scenarios (RCPs 2.6 and 8.5). Different sowing dates (shown at the top of each column) were evaluated to assess the effect of these adaptation solutions. Data are shown as scatter plots with LOESS smooth lines to show the trend. Values shown represent averages from simulations using three global climate models. **(B)** Outputs from linear regression analysis of crop cycle length over time, performed on each group of the dataset as described above.

The table (B) in the figure:

| RCP | sowing_date | location | r.squared | slope | p.value |
|---|---|---|---|---|---|
| 26 | 11May | Cahul | 0.000588 | 0.00974 | 0.709 |
| 26 | 26April | Cahul | 0.000756 | 0.0104 | 0.672 |
| 26 | 26May | Cahul | 0.00265 | 0.025 | 0.427 |
| 26 | 11May | Chisinau | 0.0014 | -0.0178 | 0.564 |
| 26 | 26April | Chisinau | 0.00135 | -0.0152 | 0.571 |
| 26 | 26May | Chisinau | 0.00121 | -0.0344 | 0.593 |
| 26 | 11May | Soroca | 0.00127 | -0.0173 | 0.583 |
| 26 | 26April | Soroca | 0.000326 | -0.00764 | 0.781 |
| 26 | 26May | Soroca | 0.000605 | -0.033 | 0.705 |
| 85 | 11May | Cahul | 0.291 | -0.24 | 1.58e-19 |
| 85 | 26April | Cahul | 0.319 | -0.235 | 1.32e-21 |
| 85 | 26May | Cahul | 0.283 | -0.267 | 6.56e-19 |
| 85 | 11May | Chisinau | 0.317 | -0.266 | 1.8e-21 |
| 85 | 26April | Chisinau | 0.337 | -0.253 | 5.03e-23 |
| 85 | 26May | Chisinau | 0.309 | -0.316 | 7.73e-21 |
| 85 | 11May | Soroca | 0.383 | -0.298 | 9.62e-27 |
| 85 | 26April | Soroca | 0.385 | -0.27 | 5.87e-27 |
| 85 | 26May | Soroca | 0.244 | -0.43 | 3.91e-16 |

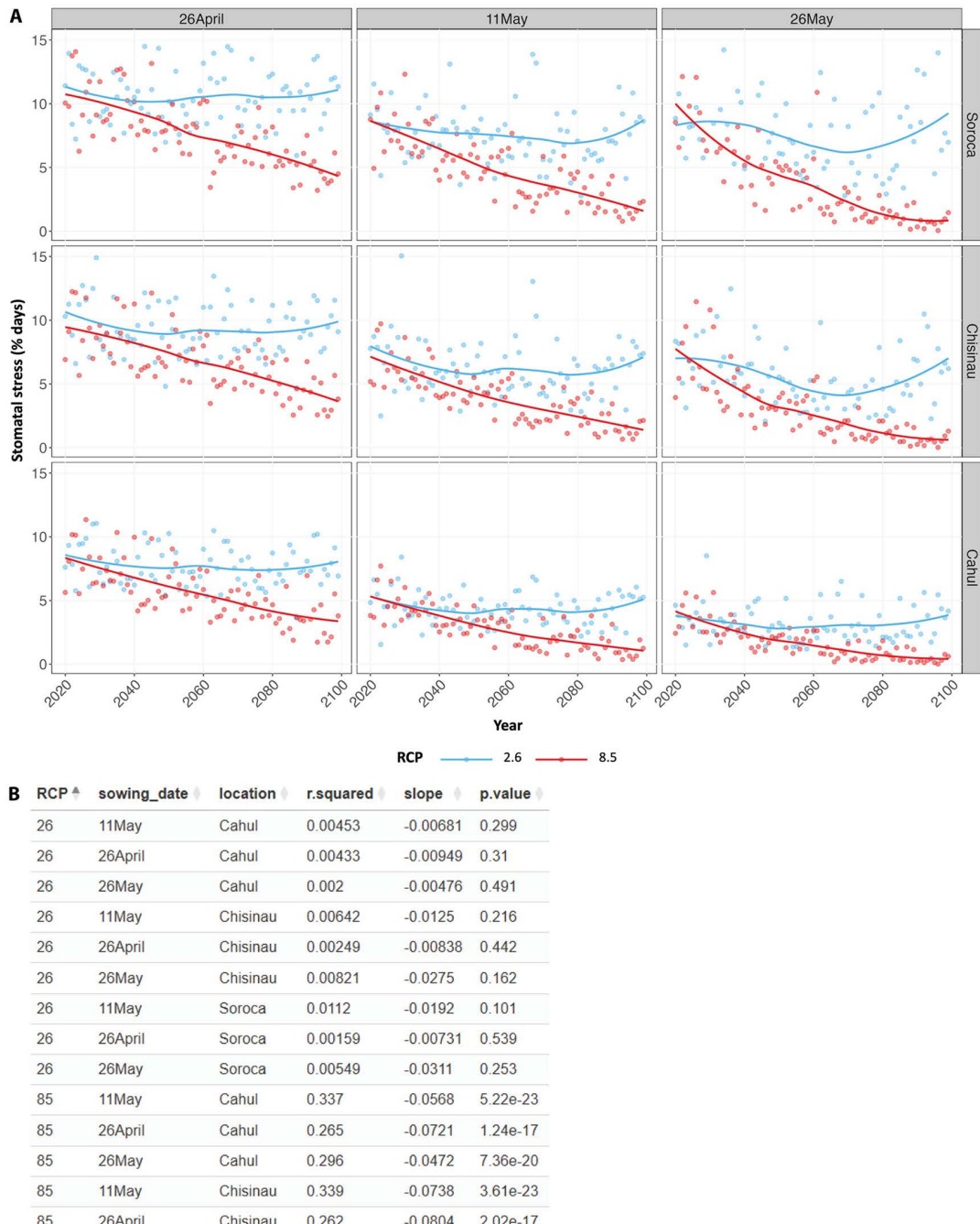

**RCP** — 2.6 — 8.5

| RCP | sowing_date | location | r.squared | slope | p.value |
|---|---|---|---|---|---|
| 26 | 11May | Cahul | 0.00453 | -0.00681 | 0.299 |
| 26 | 26April | Cahul | 0.00433 | -0.00949 | 0.31 |
| 26 | 26May | Cahul | 0.002 | -0.00476 | 0.491 |
| 26 | 11May | Chisinau | 0.00642 | -0.0125 | 0.216 |
| 26 | 26April | Chisinau | 0.00249 | -0.00838 | 0.442 |
| 26 | 26May | Chisinau | 0.00821 | -0.0275 | 0.162 |
| 26 | 11May | Soroca | 0.0112 | -0.0192 | 0.101 |
| 26 | 26April | Soroca | 0.00159 | -0.00731 | 0.539 |
| 26 | 26May | Soroca | 0.00549 | -0.0311 | 0.253 |
| 85 | 11May | Cahul | 0.337 | -0.0568 | 5.22e-23 |
| 85 | 26April | Cahul | 0.265 | -0.0721 | 1.24e-17 |
| 85 | 26May | Cahul | 0.296 | -0.0472 | 7.36e-20 |
| 85 | 11May | Chisinau | 0.339 | -0.0738 | 3.61e-23 |
| 85 | 26April | Chisinau | 0.262 | -0.0804 | 2.02e-17 |
| 85 | 26May | Chisinau | 0.284 | -0.0846 | 4.85e-19 |
| 85 | 11May | Soroca | 0.348 | -0.0892 | 6.67e-24 |
| 85 | 26April | Soroca | 0.268 | -0.0875 | 7.27e-18 |
| 85 | 26May | Soroca | 0.248 | -0.142 | 1.92e-16 |

**Fig 7. Exposure to temperature stress affecting stomatal transpiration showed similar trends as the crop cycle length. (A)** Exposure to temperature stress affecting stomatal transpiration (stomatal stress) expressed as the percentage of days crops are exposed to the stress (% days) predicted from AquaCrop simulations of maize grown at three locations in the Republic of Moldova (Soroca, Chisinau, and Cahul) over the course of the century under two climate change scenarios (RCPs 2.6 and 8.5). Different sowing dates (shown at the top of each column) were evaluated to assess the effect of these adaptation solutions. Data are shown as scatter plots with LOESS smooth lines to show the trend. Values shown represent averages from simulations using three global climate models. **(B)** Outputs from linear regression analysis of stomatal stress over time, performed on each group of the dataset as described above.

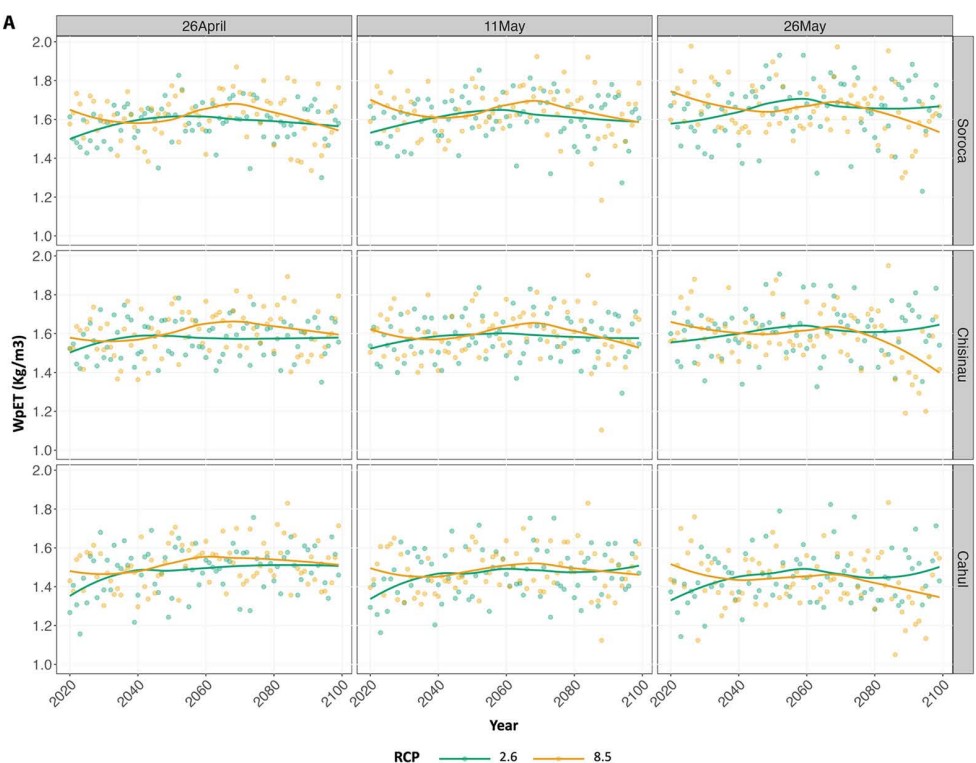

**B**

| RCP | sowing_date | time.window | WPet.mean | WPet.SD |
|---|---|---|---|---|
| 26 | 11May | 2020-2039 | 1.52 | 0.258 |
| 26 | 11May | 2040-2059 | 1.57 | 0.222 |
| 26 | 11May | 2060-2079 | 1.57 | 0.245 |
| 26 | 11May | 2080-2099 | 1.54 | 0.239 |
| 26 | 26April | 2020-2039 | 1.52 | 0.213 |
| 26 | 26April | 2040-2059 | 1.56 | 0.197 |
| 26 | 26April | 2060-2079 | 1.56 | 0.204 |
| 26 | 26April | 2080-2099 | 1.55 | 0.209 |
| 26 | 26May | 2020-2039 | 1.53 | 0.300 |
| 26 | 26May | 2040-2059 | 1.60 | 0.272 |
| 26 | 26May | 2060-2079 | 1.58 | 0.281 |
| 26 | 26May | 2080-2099 | 1.57 | 0.288 |
| 85 | 11May | 2020-2039 | 1.56 | 0.240 |
| 85 | 11May | 2040-2059 | 1.56 | 0.250 |
| 85 | 11May | 2060-2079 | 1.62 | 0.233 |
| 85 | 11May | 2080-2099 | 1.55 | 0.257 |
| 85 | 26April | 2020-2039 | 1.55 | 0.203 |
| 85 | 26April | 2040-2059 | 1.57 | 0.234 |
| 85 | 26April | 2060-2079 | 1.62 | 0.223 |
| 85 | 26April | 2080-2099 | 1.57 | 0.268 |
| 85 | 26May | 2020-2039 | 1.59 | 0.279 |
| 85 | 26May | 2040-2059 | 1.56 | 0.281 |
| 85 | 26May | 2060-2079 | 1.59 | 0.258 |
| 85 | 26May | 2080-2099 | 1.48 | 0.341 |

**Fig 8. A declining trend in water productivity was observed with delayed sowing date under RCP 8.5 climate scenario. (A)** Evapotranspiration water productivity (WpET, kg yield produced per $m^3$ water evapotranspired) predicted from AquaCrop simulations of maize grown at three locations in the Republic of Moldova (Soroca, Chisinau, and Cahul) over the course of the century under two climate change scenarios (RCPs 2.6 and 8.5). Different sowing dates (shown at the top of each column) were evaluated to assess the effect of these adaptation solutions. Data are shown as scatter plots with LOESS smooth lines to show the trend. Values shown represent averages from simulations using three global climate models. **(B)** Outputs from time window summary analysis of water productivity (WpET). The mean and standard deviation (SD) are calculated from 20 years time windows, performed on each group of the dataset as described above.

Since AquaCropPlotter was designed to support a broad range of users without extensive computational backgrounds, it provides a set of simple features for visualization and statistical analysis. Although this tool is often sufficient for gaining insights into AquaCrop data, more advanced analyses may require other tools. Nevertheless, the automated data processing step of AquaCropPlotter combines information from all the uploaded AquaCrop simulation results to produce a tidy machine-readable dataset, which can serve as a convenient starting point to facilitate further or more sophisticated analyses by other software.

## Conclusion

AquaCropPlotter is an open-source tool designed to streamline the processing, visualization and analysis of outputs from AquaCrop. Its intuitive interface enables not only easy batch upload and automated processing of large datasets but also flexible visualization functionalities and simple statistical analysis tools for investigating the data. AquaCropPlotter is a valuable complement to AquaCrop as it facilitates the generation of useful insights from the simulation results and equips users for handling complicated datasets, such as the comparison of crop yield under multiple climate scenarios or assessment of different adaptation solutions. Its adoption by a diverse range of practitioners and researchers, especially those without extensive computational expertise, is anticipated, expanding the utilization of AquaCrop and broadening the wide-ranging impact of crop model in agriculture.

## Supporting information

**S1 Table. Details of AquaCrop output variables.** A list of AquaCrop output variables with corresponding details including description, unit, output category, dataset type and associated output file.
(CSV)

## Acknowledgments

We acknowledge the support from the Food and Agriculture Organization of the United Nations (FAO) Land and Water Division (NSL) and Office of Climate Change, Biodiversity and Environment (OCB). We also thank AquaCrop extended core group of experts for their invaluable insights on how to improve AquaCropPlotter during hands-on training sessions.

## Author contributions

**Conceptualization:** Nattapong Sanguankiattichai, Riccardo Soldan.

**Data curation:** Nattapong Sanguankiattichai, Andrea Setti, Jorge Alvar-Beltrán.

**Formal analysis:** Nattapong Sanguankiattichai, Andrea Setti, Jorge Alvar-Beltrán.

**Funding acquisition:** Maher Salman, Riccardo Soldan.

**Investigation:** Nattapong Sanguankiattichai, Andrea Setti, Jorge Alvar-Beltrán, Dirk Raes, Riccardo Soldan.

**Methodology:** Nattapong Sanguankiattichai, Andrea Setti, Jorge Alvar-Beltrán, Riccardo Soldan.

**Project administration:** Nattapong Sanguankiattichai, Maher Salman, Riccardo Soldan.

**Resources:** Maher Salman.

**Software:** Nattapong Sanguankiattichai, Riccardo Soldan.

**Supervision:** Maher Salman, Riccardo Soldan.

**Validation:** Nattapong Sanguankiattichai, Andrea Setti, Jorge Alvar-Beltrán, Dirk Raes, Riccardo Soldan.

**Visualization:** Nattapong Sanguankiattichai, Andrea Setti, Jorge Alvar-Beltrán.

**Writing – original draft:** Nattapong Sanguankiattichai.

**Writing – review & editing:** Nattapong Sanguankiattichai, Andrea Setti, Jorge Alvar-Beltrán, Maher Salman, Dirk Raes, Riccardo Soldan.

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
