## [Decision Letter · Decision Letter 0]

3 Sep 2025

PLOS ONE

Dear Dr. Sanguankiattichai,

Thank you for submitting your manuscript to PLOS ONE. After careful consideration, we feel that it has merit but does not fully meet PLOS ONE’s publication criteria as it currently stands. Therefore, we invite you to submit a revised version of the manuscript that addresses the points raised during the review process.

We look forward to receiving your revised manuscript.

Kind regards,

Jamil Alexandre Ayach Anache

Academic Editor

PLOS ONE

Additional Editor Comments:

Dear authors, I received two reviews to your papers. All comments are relevant and must be adressed.

Best regards,

Academic editor

Reviewers' comments:

Reviewer's Responses to Questions

**Comments to the Author**

1. Is the manuscript technically sound, and do the data support the conclusions?

Reviewer #1: Yes

Reviewer #2: Yes

2. Has the statistical analysis been performed appropriately and rigorously?

Reviewer #1: Yes

Reviewer #2: N/A

3. Have the authors made all data underlying the findings in their manuscript fully available?

Reviewer #1: Yes

Reviewer #2: Yes

4. Is the manuscript presented in an intelligible fashion and written in standard English?

Reviewer #1: Yes

Reviewer #2: Yes

Reviewer #1: The authors have made a valuable contribution to the crop modeling and natural resource management community by developing a comprehensive set of tools for visualizing and analyzing AquaCrop outputs. These tools include a Windows GUI, a ShinyApp, and an R package. Additionally, the paper presents a case study conducted across three locations under different climate change scenarios. The manuscript is generally well-structured, clearly written, and follows a logical progression that is easy to follow. However, I have some concerns and suggestions that I provide detailed comments below.

Comment 1: In the introduction, the authors present the importance and implications of developing an application for visualizing and analysing AquaCrop, which is well-done. However, the authors should bear in mind that there will be two kinds of readers: the ones that are acquainted with AquaCrop and understand all the equations and how the physical processes are represented in the model and the ones that will be firstly introduced by AquaCrop by this paper. While I understand that the main target group can potentially be the reader that is acquainted with AquaCrop, I think that giving more details of AquaCrop is of utmost importance. This includes, but it is not limited to, details of the input data that is required and the units (maybe a table?), the main equations and physical principles, and limitations of the model.

Comment 2: The citation and reference of the packages used in to develop the R application is essential, however, it is important to explain what is the purpose of using a given package. The authors have done this by explaining that Tudyverse and furr were used for analysis pipelines. A similar explanation should be given for shinydashboard, shinyjs, shinyBS, and DT.

Comment 3: The authors have done an excellent job in providing all the relevant links for accessing all the extensive resources related to AquaCropPlotter (github and Mendeley repositories, links for download and technical documentation). All links are working correctly, nonetheless, I strongly suggest presenting them as a citation and put the links in the reference.

Comment 4: Maybe one of the most critical aspects of AquaCropPlotter is the ingestion of the data into the application. As the authors stated "AquaCropPlotter takes project files and output files from AquaCrop simulation as its input". I think the authors should provide more information on the differences of the Standard "GUI" and the stand-alone "Plug-in", e.g., in what cases each one of them are used? Also, the output files from each type of interface should be better explained. The standard GUI saves 10 files and the reader should learn in the text about what to expect to be written in each one of these files. For example, I can infer that the project file [.PRM] stores all the project settings, however, for the file CompEC.OUT I cannot make any inference if I'm not an experienced user of AquaCROP. The same reasoning applies to the outputs of the "Plug-in".

Comment 5: In the Step 4: Analysis, the authors provide an explanation that "AquaCropPlotter offers two simple statistical analysis options for gaining more insights into the data". The first one is a window summary function that is not clear to me what it is exactly. The authors are encouraged to give more details about how the summary and trend of variables are computed in the application. Moreover, they should present one example for each analysis in the case study.

Reviewer #2: The manuscript is very well written and is a substantial improvement as an option to the traditional AquaCrop model software. However, the figures need to be improved because they are of poor quality (blurry).

**Do you want your identity to be public for this peer review?** For information about this choice, including consent withdrawal, please see our Privacy Policy

Reviewer #1: **Yes: ** Marcos Roberto Benso

Reviewer #2: No

---

## [Author Response · Author response to Decision Letter 1]

18 Oct 2025

Response to reviewers

Reviewer #1: The authors have made a valuable contribution to the crop modeling and natural resource management community by developing a comprehensive set of tools for visualizing and analyzing AquaCrop outputs. These tools include a Windows GUI, a ShinyApp, and an R package. Additionally, the paper presents a case study conducted across three locations under different climate change scenarios. The manuscript is generally well-structured, clearly written, and follows a logical progression that is easy to follow. However, I have some concerns and suggestions that I provide detailed comments below.

- Comment 1: In the introduction, the authors present the importance and implications of developing an application for visualizing and analysing AquaCrop, which is well-done. However, the authors should bear in mind that there will be two kinds of readers: the ones that are acquainted with AquaCrop and understand all the equations and how the physical processes are represented in the model and the ones that will be firstly introduced by AquaCrop by this paper. While I understand that the main target group can potentially be the reader that is acquainted with AquaCrop, I think that giving more details of AquaCrop is of utmost importance. This includes, but it is not limited to, details of the input data that is required and the units (maybe a table?), the main equations and physical principles, and limitations of the model.

Response: We have added more information about AquaCrop including the principles, inputs, outputs, and limitations (Line 58 - 110). We feel that listing all the inputs and their units might be quite detailed for the purpose of this manuscript, since there are many inputs that are not straightforward and there can be multiple units for a single variable, e.g. for the climate inputs. We think that providing the concepts and summary of the required inputs may be sufficient for understanding AquaCrop within the scope of this manuscript. More details of inputs can be found better explained in detail in the original AquaCrop publications cited.

- Comment 2: The citation and reference of the packages used in to develop the R application is essential, however, it is important to explain what is the purpose of using a given package. The authors have done this by explaining that Tudyverse and furr were used for analysis pipelines. A similar explanation should be given for shinydashboard, shinyjs, shinyBS, and DT.

Response: We have added more explanation for the packages used (Line 158 - 164).

- Comment 3: The authors have done an excellent job in providing all the relevant links for accessing all the extensive resources related to AquaCropPlotter (github and Mendeley repositories, links for download and technical documentation). All links are working correctly, nonetheless, I strongly suggest presenting them as a citation and put the links in the reference.

Response: We have put the links in the references and cited them in the text (Line 166-178).

- Comment 4: Maybe one of the most critical aspects of AquaCropPlotter is the ingestion of the data into the application. As the authors stated "AquaCropPlotter takes project files and output files from AquaCrop simulation as its input". I think the authors should provide more information on the differences of the Standard "GUI" and the stand-alone "Plug-in", e.g., in what cases each one of them are used? Also, the output files from each type of interface should be better explained. The standard GUI saves 10 files and the reader should learn in the text about what to expect to be written in each one of these files. For example, I can infer that the project file [.PRM] stores all the project settings, however, for the file CompEC.OUT I cannot make any inference if I'm not an experienced user of AquaCROP. The same reasoning applies to the outputs of the "Plug-in".

Response: We have added more information about

• the differences of the standard GUI and plug-in versions (Line 116 - 121)

• information on each of the output files from the standard GUI and plug-in versions (Line 190-222)

• details of output files and variables are also listed in a table provided as supplementary material Table S1.

- Comment 5: In the Step 4: Analysis, the authors provide an explanation that "AquaCropPlotter offers two simple statistical analysis options for gaining more insights into the data". The first one is a window summary function that is not clear to me what it is exactly. The authors are encouraged to give more details about how the summary and trend of variables are computed in the application. Moreover, they should present one example for each analysis in the case study.

Response: We have added more information about the analysis functions (Line 276-292) and include an example in the case study (Line 338-344, 362-365, 380-383).

Reviewer #2: The manuscript is very well written and is a substantial improvement as an option to the traditional AquaCrop model software. However, the figures need to be improved because they are of poor quality (blurry).

Response: The figures seem to be blurry when the submission system auto-generates the pdf file of the manuscript for reviewers. We have double checked our figure files and make sure the final version uploaded is not blurry.

---

## [Decision Letter · Decision Letter 1]

12 Nov 2025

AquaCropPlotter: a Shiny app for visualizing and analyzing AquaCrop simulation results

PONE-D-25-32827R1

Dear Dr. Sanguankiattichai,

We’re pleased to inform you that your manuscript has been judged scientifically suitable for publication and will be formally accepted for publication once it meets all outstanding technical requirements.

Kind regards,

Jamil Alexandre Ayach Anache

Academic Editor

PLOS ONE

Additional Editor Comments (optional):

Reviewers' comments:

Reviewer's Responses to Questions

**Comments to the Author**

Reviewer #1: All comments have been addressed

2. Is the manuscript technically sound, and do the data support the conclusions?

Reviewer #1: Yes

3. Has the statistical analysis been performed appropriately and rigorously?

Reviewer #1: Yes

4. Have the authors made all data underlying the findings in their manuscript fully available?

Reviewer #1: Yes

5. Is the manuscript presented in an intelligible fashion and written in standard English?

Reviewer #1: Yes

Reviewer #1: I appreciate the effort made by the authors in responding to the comments and suggestions I have made. After carefully evaluating the author's response, I believe that the manuscript is ready to be published in PLOS One.

I would like to thank the authors for creating a useful and interesting tool for crop modeling, which I believe will be greatly appreciated by the research and technical community. Moreover, I truly believe that the readership of PLOS One will be interested in reading the manuscript.

Regards

**Do you want your identity to be public for this peer review?** For information about this choice, including consent withdrawal, please see our Privacy Policy

Reviewer #1: **Yes: ** Marcos Roberto Benso

---

## [Editor Report · Acceptance letter]

PONE-D-25-32827R1

PLOS ONE

Dear Dr. Sanguankiattichai,

I'm pleased to inform you that your manuscript has been deemed suitable for publication in PLOS ONE. Congratulations! Your manuscript is now being handed over to our production team.

Kind regards,

on behalf of

Dr. Jamil Alexandre Ayach Anache

Academic Editor

PLOS ONE